# Notch signaling regulates vasculogenic mimicry and promotes cell morphogenesis and the epithelial-to-mesenchymal transition in pancreatic ductal adenocarcinoma

Nontawat Benjakul[1,2], Nattapa Prakobphol[3], Chayada Tangshewinsirikul[4☯], Wirada Dulyaphat[4☯], Jisnuson Svasti[3,5], Komgrid Charngkaew[1‡], Thaned Kangsamaksin[3‡*]

1 Faculty of Medicine Siriraj Hospital, Department of Pathology, Mahidol University, Bangkok, Thailand, 2 Faculty of Medicine Vajira Hospital, Department of Anatomical Pathology, Navamindradhiraj University, Bangkok, Thailand, 3 Faculty of Science, Department of Biochemistry, Mahidol University, Bangkok, Thailand, 4 Faculty of Medicine Ramathibodi Hospital, Division of Maternal Fetal Medicine, Department of Obstetrics and Gynecology, Mahidol University, Bangkok, Thailand, 5 Laboratory of Biochemistry, Chulabhorn Research Institute, Bangkok, Thailand

☯ These authors contributed equally to this work.
‡ These authors also contributed equally to this work.
* thaned.kan@mahidol.ac.th

## Abstract

Vasculogenic mimicry (VM) is the process where cancer cells adopt endothelial characteristics by forming tube-like structures and perfusing channels. This phenomenon has been demonstrated in several types of solid tumors and associated with the growth and survival of tumor cells. In this study, we investigated the presence of VM formation in human pancreatic ductal adenocarcinoma (PDAC) and elucidated the molecular mechanisms underlying the VM process. In human PDAC tissues, CD31-negative, periodic acid-Schiff (PAS)-positive channels were predominantly found in desmoplastic areas, which are generally also hypovascularized. We found a positive correlation of VM capacity to tumor size and NOTCH1 expression and nuclear localization with statistical significance, implicating that Notch activity is involved with VM formation. Additionally, our data showed that the presence of growth or angiogenic factors significantly increased Notch activity in PDAC cell lines and upregulated several mesenchymal marker genes, such as *TWIST1* and *SNAI1*, which can be inhibited by a gamma-secretase inhibitor. Our data showed that Notch signaling plays an important role in inducing VM formation in PDAC by promoting the epithelial-to-mesenchymal transition process.

## Introduction

Pancreatic cancer is among the most lethal human cancer with an average 5-year survival rate of less than 5% [1]. There are a number of types of pancreatic cancer, but pancreatic ductal

**Data Availability Statement:** All relevant data are within the paper and its Supporting Information files.

**Funding:** This research has received funding support from the NSRF via the Program Management Unit for Human Resources & Institutional Development, Research and Innovation (Grant Number B05F640133), National Research Council of Thailand and Mahidol University (Grant Number NRCT5-RSA63015-11) (T.K.) and the Department of Pathology, Faculty of Medicine Siriraj Hospital, Mahidol University (K.C.). The details of the funders can be found at (1) https://www.nxpo.or.th/B, (2) https://www.nrct.go.th, and (3) https://www.mahidol.ac.th. The funders had no role in study design, data collection and analysis, decision to publish, or preparation of the manuscript.

**Competing interests:** The authors have declared that no competing interests exist.

adenocarcinoma (PDAC) is the most common and accounts for about 85% of cases. Surgical resection is the only cure, which can improve the 5-year survival rate up to 20% [2]. However, most pancreatic cancer patients are often diagnosed at an advanced stage, which renders treatable surgery impossible. In addition, pancreatic cancer has been shown to be insensitive to many chemotherapeutic drugs. The current standard-of-care therapy improves patient survival by only a matter of weeks [3].

It is widely accepted that the growth of solid tumors requires a continuous supply of oxygen and nutrients [3,4]. One of the most important factors and characteristic hallmarks for cancer development and progression is the angiogenic switch. Therefore, genetic manipulations and pharmacological perturbations of angiogenesis have been extensively investigated, and data from various studies have led to the development of a number of cancer therapeutics targeting angiogenic processes [5]. Bevacizumab (Avastin, Genentech/Roche), one of the first clinically-approved angiogenesis inhibitors, has been used to treat patients with metastatic colorectal, lung, renal, and ovarian cancers by blocking the vascular endothelial growth factor (VEGF) pathway [6]. However, after the initial response phase from anti-angiogenic treatment, tumors often begin to resurface and the disease progression resumes. Increasing evidence demonstrates that tumor cells develop an adaptive response and become resistant to angiogenesis inhibitors. There are several proposed mechanisms of how a tumor can evade angiogenesis inhibition, including upregulation of alternative pro-angiogenic signaling pathways, increased protection of tumor vasculature from anti-angiogenic drugs, and vasculogenic mimicry [5–8].

Vasculogenic mimicry (VM) has been characterized by the ability of cancer cells to express endothelium-associated genes and generate extracellular matrix (ECM)-rich vascular networks, which recapitulate embryonic vasculogenesis. This process has been associated with human aggressive tumors, including metastatic melanoma cells [9]. VM has been observed in other types of cancer, such as carcinomas (breast, ovary, lung, prostate, bladder, kidney); sarcomas (Ewing, mesothelial, synovial, osteosarcoma, alveolar rhabdomyosarcoma); and gliomas, glioblastoma, and astrocytoma [8]. The term "vascular mimicry" has been synonymously and interchangeably used with vasculogenic mimicry but may include other vascular cell-associated characteristics such as macrophages and lymphocytes.

Pancreatic carcinomas have been reported to be insensitive to the clinically used drug gemcitabine, and the tumors appear largely desmoplastic with poorly perfused and low vascularization. Several lines of evidence suggested that PDAC hypovascularity was actively induced by various factors, including interstitial fluid pressure (IFP), solid stress, and endothelial ablation although the molecular mechanisms underlying these processes were not completely understood [10–13]. These hypovascular characteristics have also been observed in a mouse model of pancreatic cancer, the KPC mice which conditionally express endogenous mutant Kras and p53 alleles in pancreatic cells [14]. Pancreatic tumors from these mice exhibited a dysfunctional vasculature. Moreover, drug delivery to KPC pancreatic tumors was markedly inefficient when compared with adjacent tissues or other transplanted tumors [14]. Previous reports showed that no improved survival was observed in advanced PDAC patients under gemcitabine and bevacizumab treatment [15]. These data confirmed that angiogenesis may not represent a predominant factor to promote the growth of pancreatic cancer, and there are likely alternative mechanisms for pancreatic cancer cells to obtain oxygen and nutrients in order to support their uncontrolled growth and metastasis. In this study, we determined the presence of vasculogenic mimicry in PDAC cases in Thailand and found that the incidence was often associated with desmoplastic areas of PDAC and significantly correlated with Notch activity. Additionally, we found that PDAC cell lines when activated by growth or angiogenic factors were able to undergo endothelial-like cell morphogenesis in vitro via a Notch-dependent pathway. Overall, our study demonstrates that Notch signaling plays an essential role in regulating

the process of vasculogenic mimicry in PDAC by promoting the epithelial-to-mesenchymal transition and endothelial characteristics.

## Materials and methods

### Cells and reagents

All cell cultures were maintained at 37˚C in a mixture of 5% $CO_2$ and 95% humidified air. PANC-1 and MIA-PaCa-2 cell lines were obtained from the American Type Culture Collection (ATCC, USA). Both cancer cell lines were maintained in 1X High-Glucose DMEM (Invitrogen) with 10% fetal bovine serum (FBS) and 1X Pen-Strep (Invitrogen). The collection of umbilical cords and human umbilical vein endothelial cells (HUVECs) in this study was approved by the Faculty of Medicine Ramathibodi Hospital Human Research Ethics Committee, Mahidol University (MURA2015/344). HUVECs were isolated as described [16] and grown in Endothelial Cell Growth Media-2 media (EGM-2, Lonza) on a culture plate coated with rat tail type I collagen (BD Biosciences).

### Network formation assay

Porcine Collagen (Wako) was prepared on ice. The collagen mixture (Porcine Collagen: 10X RPMI 1640 medium: sterile water: Neutralizing Buffer; 7:1:1:1) was added to 24-well plates and incubated at 37˚C for 1 hour to solidify and form collagen gel. Cells were then overlaid at $1.0 \times 10^5$ cells per well in Human Endothelial Serum Free Medium (SFM, Invitrogen) supplemented with 20 ng/ml EGF (Invitrogen) or rhVEGF-$A_{165}$ (R&D). After 2 hours when the cells completely adhered, the medium was removed, and another layer of collagen gel was added to the plates and incubated at 37˚C for 3 hours. Then, complete medium with compounds was added to each well and changed every other day. Network formation and branching were photographed every day for 4 days.

### Quantitative real-time PCR (qRT-PCR)

Total RNA was collected using the GF-1 Total RNA Extraction kit (Vivantis), and cDNA was synthesized with the RevertAid Reverse Transcriptase (Thermo Scientific). PCR reaction was performed using Luna Universal qPCR Master Mix (New England Biolabs). Forward and reverse primers for qRT-PCR (5'–3') were: *NOTCH1*, CTCACCTGGTGCAGACCCAG and GCACCTGTAGCTGGTGGCTG; *HEY1*, ACGAGAATGGAAACTTGAGTTC and AACTCCGATA GTCCATAGCAAG; *HES1*, CCCAACGCAGTGTCACCTTC and TACAAAGGCGCAATCCAATA TG; *NRARP*, GGGCTGCATAGAAAATTGGA and CCCTTTTTAGCCTCCCAGAG; *CDH1*, GCC TCCTGAAAAGAGAGTGGAAG and TGGCAGTGTCTCTCCAAATCCG; *CDH2*, CCTCCAGAGT TTACTGCCATGAC and GTAGGATCTCCGCCACTGATTC; *CDH5*, TGTGACAGCAGTGGATG CAGA and CTGTACTTGGTCATCCGGTTCTG; *VIM*, AGGCAAAGCAGGAGTCCACTGA and ATCTGGCGTTCCAGGGACTCAT; *TWIST1*, GCCAGGTACATCGACTTCCTCT and TCCATC CTCCAGACCGAGAAGG; *SNAI1*; TGCCCTCAAGATGCACATCCGA and GGGACAGGAGAAGG GCTTCTC; *ACTB*, CGAGGCCCAGAGCAAGAGAG and CTCGTAGATGGGCACAGTGTG.

### Tissue preparation

This study was approved by the Siriraj Institutional Review Board (Si 014/2017). In this study, 122 formalin-fixed, paraffin-embedded human PDAC tissue samples were obtained between January, 2008 and December, 2016. The written consent was obtained from participants prior to tissue analysis. All patients did not receive chemotherapy or radiotherapy before operation. All diagnoses were confirmed by clinical examination and histological confirmation of

hematoxylin and eosin (H&E), Periodic-Schiff Acid (PAS), immunostaining with a CD31 monoclonal antibody (JC/70A, Thermo Fisher Scientific), and a NOTCH1 polyclonal antibody (Abcam), and data were analyzed by clinical pathologists (N.B. and K.C.).

## Statistical analysis

The SPSS23.0 software was used for statistical analysis. The associations between VM and clinicopathologic parameters and the differential expression of NOTCH1 between groups were compared using Chi-squared or Fisher's exact tests. Differences between groups were assessed by the Mann-Whitney U-test and Student's t-test. For in vitro experiments, all data are presented as means ± S.D. and were analyzed using one-way ANOVA and Student's t-test.

## Results and discussion

### PDAC hypovascularity was associated with desmoplasia in Thai patients

The tissues of 122 PDAC patients were paraffin-embedded and sectioned for H&E and PAS staining as well as CD31 immunohistochemistry for blood vessel assessment. We observed that CD31-positive blood vessels were sparsely and unevenly scattered throughout tumor sections (Fig 1A). Some of these vascular channels contained a group of red blood cells, indicating that they were functional vessels. Similar to previous reports, our PDAC sections exhibited large desmoplastic tumor areas (DM) with intense PAS staining of connective tissues and stromal components (Fig 1B). This desmoplastic growth was also characterized by low cellularity with minimal and disorganized blood vessel infiltration (Fig 1B–1D), which was significantly distinct from the adjacent vascularized, non-desmoplastic areas (non-DM). These results confirmed that CD31-positive vessel growth was inversely correlated with the level of desmoplasia in PDAC tumors.

### Association of VM frequency with clinicopathological features of PDAC cases

Using H&E, CD31 and PAS staining, VM was distinguished by channels or tube-like structures lined with pancreatic cancer cells instead of shuttle-like endothelial cells. VM channels showed a positive staining for PAS but a negative expression for CD31, confirming that cells around the channels were not composed of endothelium (Fig 2A–2D). Red blood cells (RBC) were also found inside the VM channels (Fig 2A and 2C). No necrotic and infiltrating inflammatory cells were observed around the channels. VM was detected in 91 (74.6%) of 122 pancreatic cancer cases. The clinical and pathological features of all 122 pancreatic cancer cases were summarized in Table 1. The presence of VM was correlated with tumor size (p < 0.021). The frequency of VM was significantly higher in T2 (size > 2 and ≤ 4 cm) (66/82, 80.5%) than in T3 (size > 4 cm) (15/25, 72.0%) and T1 (size ≤ 2 cm) (7/15, 46.7%). VM was observed in 4/8 (50%) of well differentiated tumors, in 80/107 (74.8%) of moderately differentiated tumors and in 7/7 (100%) of poorly differentiated tumors. No significant correlations were found between VM and patient age, gender, tumor location, perineural or angiolymphatic invasion, and AJCC 8[th] prognostic stage group.

### PDAC cells lining VM channels showed strong NOTCH1 expression and activity

Notch signaling has been implicated in regulating pancreatic cell differentiation between the endocrine and progenitor fates in the developing pancreas [17] and its involvement in the development and progression of PDAC has been demonstrated [18–22]. Therefore, we

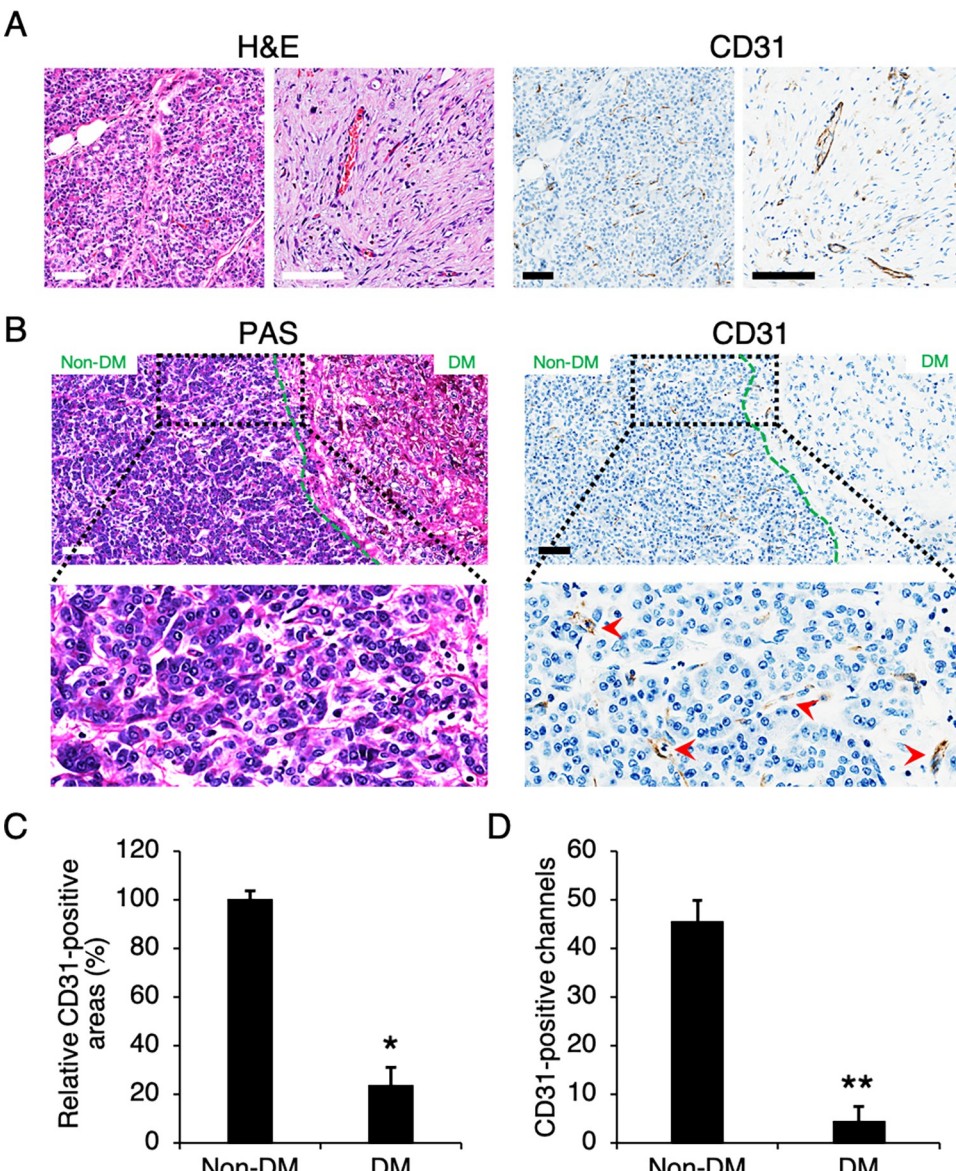

**Fig 1. PDAC hypovascularity was associated with desmoplasia.** (A) PDAC tissues were stained with H&E and immunostained for CD31. (B) PDAC tissues were stained with PAS and immunostained for CD31, and the border between non-desmoplastic (Non-DM) and desmoplastic (DM) regions were photographed and analyzed for (C) CD31-positive areas and (D) CD31-positive channels. Red arrows, CD31-positive cells. Scale bars, 200 μm. Data presented ± S.D. *P value < 0.01, **P value < 0.001 (n = 4–5).

investigated the association between VM incidence and NOTCH1. We found that VM-containing tumors showed the higher intensity and greater areas of NOTCH1 expression when compared with the non-VM group (75.8% vs. 24.2%, p < 0.01) (Fig 3A, Table 2). In addition, NOTCH1 was predominantly expressed with moderate intensity in the cytoplasm and nucleus or nucleus alone, whereas the signals were weakly distributed in the cytoplasm in the non-VM group (Fig 3B, Table 2), implicating that PDAC cells surrounding VM channels exhibited more active Notch signaling. Our results demonstrated that VM was associated with NOTCH1 expression in PDAC, and the activation of NOTCH1 may elicit changes in pancreatic cancer cell morphology or allow the cancer cells to adopt endothelial-like characteristics.

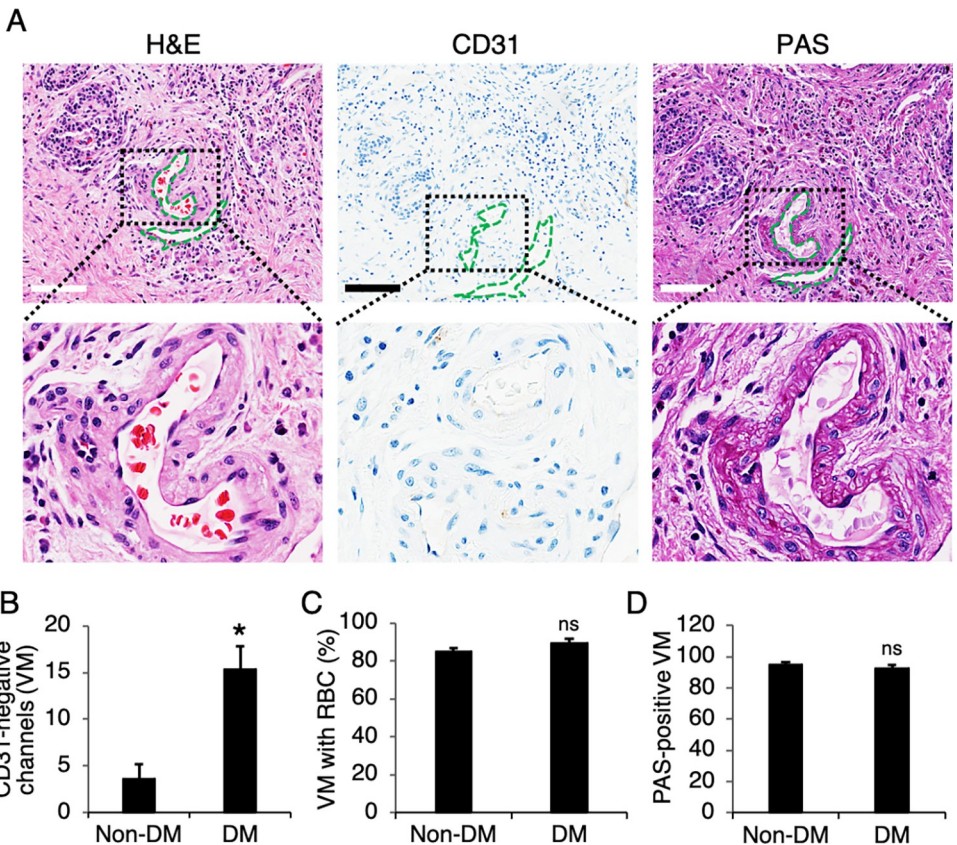

**Fig 2. CD31-negative, PAS-negative channels were predominantly found in desmoplastic areas of PDAC.** (A) PDAC tissues were stained with H&E, PAS and immunostained for CD31, and the insets demonstrated the vessel-like structures containing RBC. (B) CD31-negative channels or VM structures, (C) VM with RBC, and PAS-positive VM were quantitated and analyzed. Scale bars, 200 μm. Data presented ± S.D. *P value < 0.01 (n = 4–5).

## PDAC cells possess the abilities to undergo cell morphogenesis into tube-like structures in vitro

Next, we investigated whether PDAC cell lines could be employed as an in vitro model to study VM formation and the underlying mechanisms. We selected the two well-established PDAC cells, PANC-1 and MIA-PaCa-2, as the representative cell lines and found that, when cultured on an adherent substrate, these cells appeared cobble-like and proliferated in clusters. Interestingly, when a growth factor (EGF) or angiogenic factor (VEGF) was added to the culture media, both cells exhibited changes in cell morphology, including spindle-like shape with noticeable protruding filopodia (Fig 4A). The number of cells with filopodia was about 40% greater in cells treated with VEGF than EGF (Fig 4B). We observed similar changes when these cells were cultured between two collagen layers in the endothelial network formation assay. This assay normally allows endothelial cells to undergo cell morphogenesis and develop into network-like or tube-like structures with 1–4 days of culture in the presence of EGF and VEGF [23]. We found that PANC-1 cells began to form strings of cells and network-like clusters after two days of culture while MIA-PaCa-2 cells adopted similar cell morphology but to a much lesser degree (Fig 4C–4E) as seen in the number of branching points and tubes. Previous studies have shown that endothelium-associated genes such as VE-cadherin, EphA2, CD31, and CD34 may be involved in both angiogenic and vasculogenic mimicry processes in other

**Table 1. Association of VM and PDAC parameters.**

| Parameters | Total (%) | Tissue Samples | | p Value |
|---|---|---|---|---|
| | | VM (%) | Non VM (%) | |
| **Age** | | | | |
| < 60 | 39 (32) | 28 (71.8) | 11 (28.2) | 0.627 |
| ≥ 60 | 83 (68) | 63 (75.9) | 20 (24.1) | |
| **Gender** | | | | |
| Male | 59 (48.4) | 46 (78) | 13 (22) | 0.407 |
| Female | 63 (51.6) | 45 (71.4) | 18 (28.6) | |
| **Tumor Size (cm)** | | | | |
| ≤ 2 [T1] | 15 (12.3) | 7 (46.7) | 8 (53.3) | 0.021* |
| > 2 and ≤ 4 [T2] | 82 (67.2) | 66 (80.5) | 16 (19.5) | |
| > 4 [T3] | 25 (20.5) | 18 (72.0) | 7 (28.0) | |
| **Histological Differentiation** | | | | |
| Well differentiated | 8 (6.6) | 4 (50.0) | 4 (50.0) | 0.085 |
| Moderately differentiated | 107 (87.7) | 80 (74.8) | 27 (25.2) | |
| Poorly differentiated | 7 (5.7) | 7 (100) | 0 (0) | |
| **Perineural Invasion** | | | | |
| Negative | 15 (12.3) | 9 (60.0) | 6 (40.0) | 0.206 |
| Positive | 107 (87.7) | 82 (76.6) | 25 (23.4) | |
| **Angio-Lymphatic Invasion** | | | | |
| Negative | 66 (54.1) | 48 (72.7) | 18 (27.3) | 0.608 |
| Positive | 56 (45.9) | 43 (76.8) | 13 (23.2) | |
| **AJCC Prognostic Stage** | | | | |
| 1A | 3 (2.5) | 1 (33.3) | 2 (66.7) | 0.229 |
| 1B | 36 (29.5) | 27 (75.0) | 9 (25.0) | |
| 2A | 9 (7.4) | 5 (55.6) | 4 (44.4) | |
| 2B | 43 (35.2) | 35 (81.4) | 8 (18.6) | |
| 3 | 27 (22.1) | 21 (77.8) | 6 (22.2) | |
| 4 | 4 (3.3) | 2 (50.0) | 2 (50.0) | |

tumor types [24,25]. Our PDAC cells expressed significantly lower levels of VE-cadherin (*CDH5*) and no CD31 when compared with HUVEC (Fig 4F and 4G). Our data showed that *CDH5* was significantly upregulated in PANC-1 cultured with EGF or VEGF whereas other endothelial-specific genes were not affected (Figs 4F, 4G and S1), suggesting that PANC-1 did not entirely mimic endothelial cell characteristics and VE-cadherin may play an important role in cell morphogenesis in the VM formation.

## Notch signaling was activated upon EGF and VEGF treatment, which promoted the epithelial-to-mesenchymal transition and VM morphogenesis in PDAC cells

Consistent with our immunohistochemical data of PDAC tissues, we confirmed that EGF and VEGF significantly induced *NOTCH1* expression and several Notch downstream targets, including *HEY1*, *HES1*, and *NRARP* in PANC-1 cells (Figs 5A–5C and S2). Next, due to the distinct morphological changes of PANC-1 cells such as spindle shape and filopodia formation, which are characteristic of mesenchymal-like cells, we determined the expression profile of both epithelium-associated gene (E-cadherin; *CDH1*) and mesenchymal genes (N-cadherin; *CDH2*, Vimentin; *VIM*, Twist1; *TWIST1*, and Snail; *SNAI1*) to investigate the involvement of

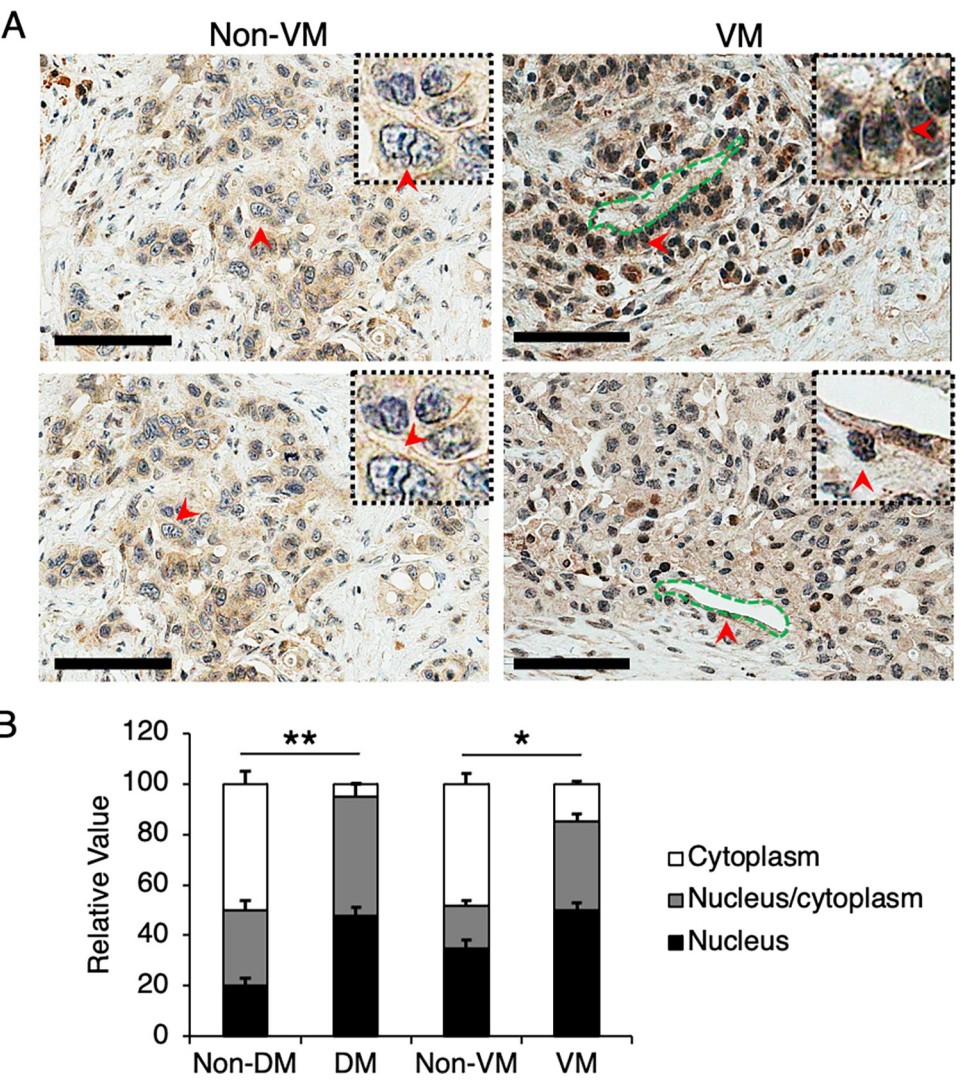

**Fig 3. PDAC cells lining VM channels showed strong NOTCH1 expression and activity.** (A) PDAC tissues were immunostained for NOTCH1 expression and localization in both non-VM and VM tissue samples. (B) NOTCH1 were quantitated based on the localization of the signals: Cytoplasm, nucleus/cytoplasm, and nucleus. Red arrows, NOTCH1-positive nuclei. Scale bars, 200 μm. Data presented ± S.D. *P value < 0.01, **P value < 0.001 (n = 4–5).

the epithelial-to-mesenchymal transition (EMT) process in the VM morphogenesis (Fig 5D). Our results demonstrated that *CDH1* was significantly downregulated (60% and 50% reduction with EGF and VEGF, respectively) while *CDH2*, *VIM*, *TWIST1*, and *SNAI1* were all upregulated (at least 1.5-fold with both EGF and VEGF), strongly suggesting that the EMT process was induced in PANC-1 cells under growth-promoting or angiogenic condition. Several lines of evidence demonstrated the importance of Notch signaling in EMT (Ref). We examined the role of Notch in our study using the gamma-secretase inhibitor dibenzazepine or DBZ to inhibit Notch signaling in our in vitro experiments. DBZ at 1 nM effectively abolished the upregulation of Notch target genes in PANC-1 cells with no significant impact on cell viability (S3 Fig). Upon DBZ treatment for 24 hours, the expression of *TWIST1* and *SNAI1* were significantly reduced by at least 40% (Figs 5E and S4). Also, the morphology of PANC-1 cells under EGF or VEGF conditions appeared round and aggregated to form a cluster rather than

Table 2. Association of VM and NOTCH1 expression and localization in PDAC.

| Parameters | Total (%) | Tissue Samples | | p Value |
|---|---|---|---|---|
| | | VM (%) | Non VM (%) | |
| **Intensity of NOTCH1 Expression** | | | | |
| Unstained | 1 (0.8) | 1 (100) | 0 (0) | < 0.01* |
| Weak | 27 (22.2) | 5 (18.5) | 22 (81.5) | |
| Moderate | 93 (76.2) | 84 (90.3) | 9 (9.7) | |
| Strong | 1 (0.8) | 1 (100) | 0 (0) | |
| **Areas of NOTCH1 Expression** | | | | |
| Only Cytoplasm | 26 (21.3) | 5 (19.2) | 21 (80.8) | < 0.01* |
| Cytoplasm > Nucleus | 76 (62.3) | 67 (88.2) | 9 (11.8) | |
| Nucleus > Cytoplasm | 20 (16.4) | 19 (95.0) | 1 (5.0) | |
| Only Nucleus | 0 (0) | 0 (0) | 0 (0) | |

spindle-shaped, network-forming structure (Fig 5F), and the quantitated numbers of cells with filopodia and tubes were reversed in the DBZ treatment group when compared to the control groups (Figs 5G and S5). Collectively, our data demonstrated that Notch signaling plays an essential role in VM morphogenesis by inducing the mesenchymal phenotype in pancreatic cancer cells, and inhibition of Notch signaling may represent a treatment strategy for PDAC by disrupting VM and inhibiting growth and metastasis of pancreatic cancer cells.

## Conclusions

Here, our findings provide evidence that the presence of VM is strongly associated with desmoplastic and hypovascularized nature of human PDAC, and vessel-like structures that are made up of cancer cells can provide a route for blood cells and nutrients for tumor survival and growth. We confirmed the development of vessel-like networks as evidence for VM by PAS positivity, CD31 negativity, and traces of red blood cells. Our clinical data revealed a positive correlation between VM and larger tumor size, implicating the role in VM in supporting the survival and growth of tumor cells especially in a larger tumor mass. Although histological differentiation did not show any significant correlation to VM formation, our statistical analysis clearly indicates a trend towards VM association with the less differentiated tumors. Unlike angiogenesis, VM depends on the morphological changes in cancer cells rather than endothelial cells to generate channels or tube-like structures that our study demonstrated could function in a similar fashion to vessel perfusion. Interestingly, several lines of evidence indicate that the process of VM formation resembles steps in angiogenesis, including cell migration, extracellular matrix remodeling, and tubulogenesis and may be induced by certain growth or angiogenic factors. This phenomenon implies that some endothelial characteristics can be activated or programmed in cancer cells via specific signal transduction pathways that are also employed during developmental vasculogenesis and angiogenesis. VE-cadherin was the only endothelial-specific gene expressed in both PDAC and HUVEC, implicating the innate ability of cancer cells to interact with one another and form tube-like structures. It is well established that VE-cadherin plays an essential role in regulating endothelial permeability and angiogenesis and its phosphorylation leads to destabilization of the adherens junction and increased monolayer permeability [26]. VEGF induces VE-caherin phosphorylation on tyrosine residues Y658 and Y731, which results in the dissolution of the cell-cell contacts [27] by disrupting the association of VE-cadherin with p120-catenin or b-catenin [28]. Hendrix and colleagues demonstrated that VE-cadherin was necessary for VM formation in melanoma cells [29]. In our study, EGF and VEGF significantly upregulated VE-cadherin expression and promoted cell

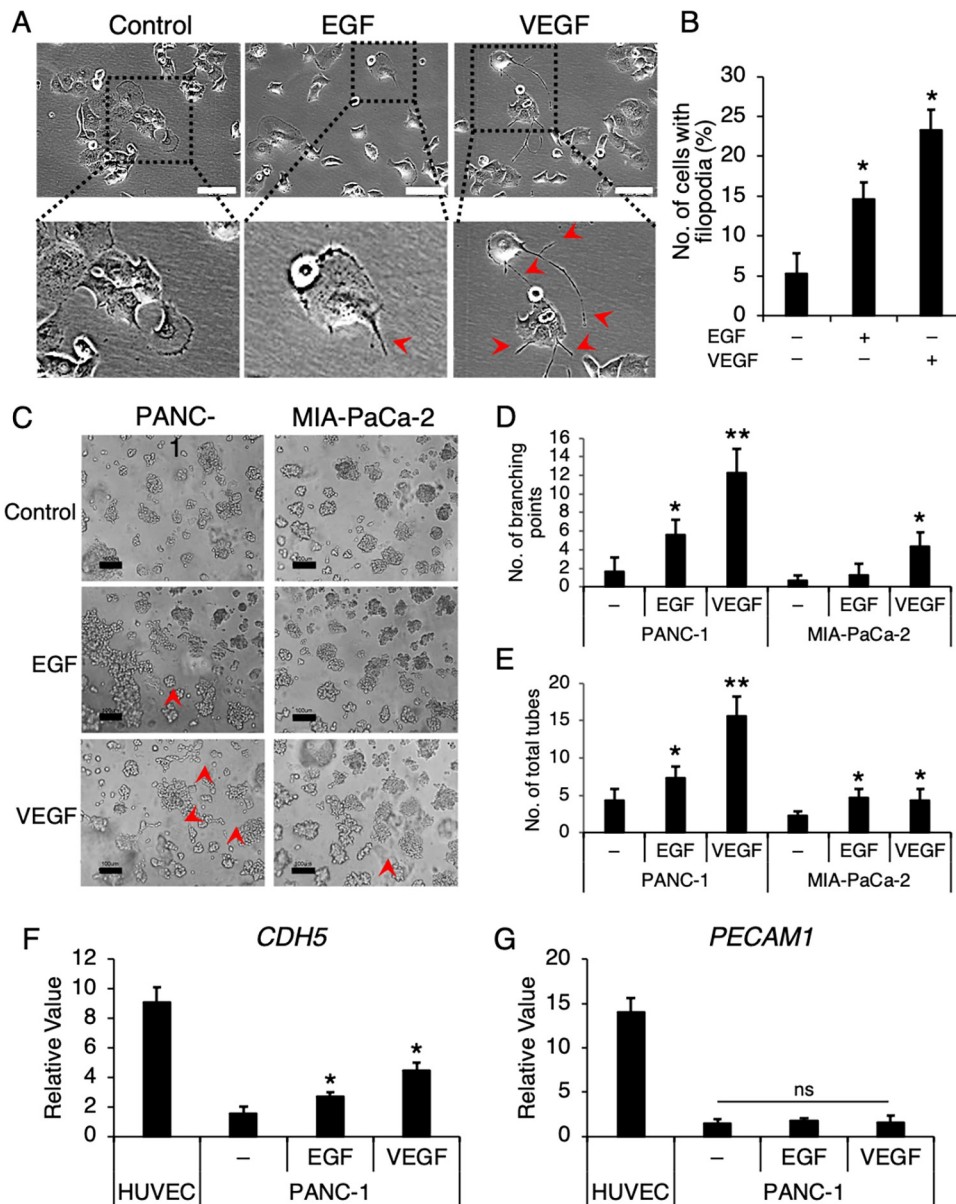

**Fig 4. EGF and VEGF induced the formation of filopodia and tube-like structures of PDAC cells in vitro.** (A) PANC-1 cells were cultured in the presence of 20 ng/ml EGF or VEGF, and their morphology was observed and (B) analyzed for number of cells with filopodia. Red arrows, filopodia-containing cells. (C) PANC-1 and MIA-PaCa-2 cells were seeded in network formation assays in the presence of EGF or VEGF and (D) analyzed for branching points and total tubes. (F) The expression of endothelial markers, *CDH5* and *PECAM1*, were determined by quantitative RT-PCR. Red arrows, filopodia and branching points. Scale bars, 100 μm. Data presented ± S.D. *P value < 0.01, **P value < 0.001, ns = not statistically significant (n = 4–5).

morphogenesis, suggesting that VE-cadherin is a cell adhesion molecule necessary for VM formation in PDAC.

In addition, our immunohistochemical analysis as well as in vitro data suggest that VM-deficient tumors and VM-low cell lines, like MIA-PaCa-2, may lack some critical factors that are required for the activation of endothelial-like capacity and VM formation. We found that the baseline expression of Notch receptors and Notch downstream targets were significantly

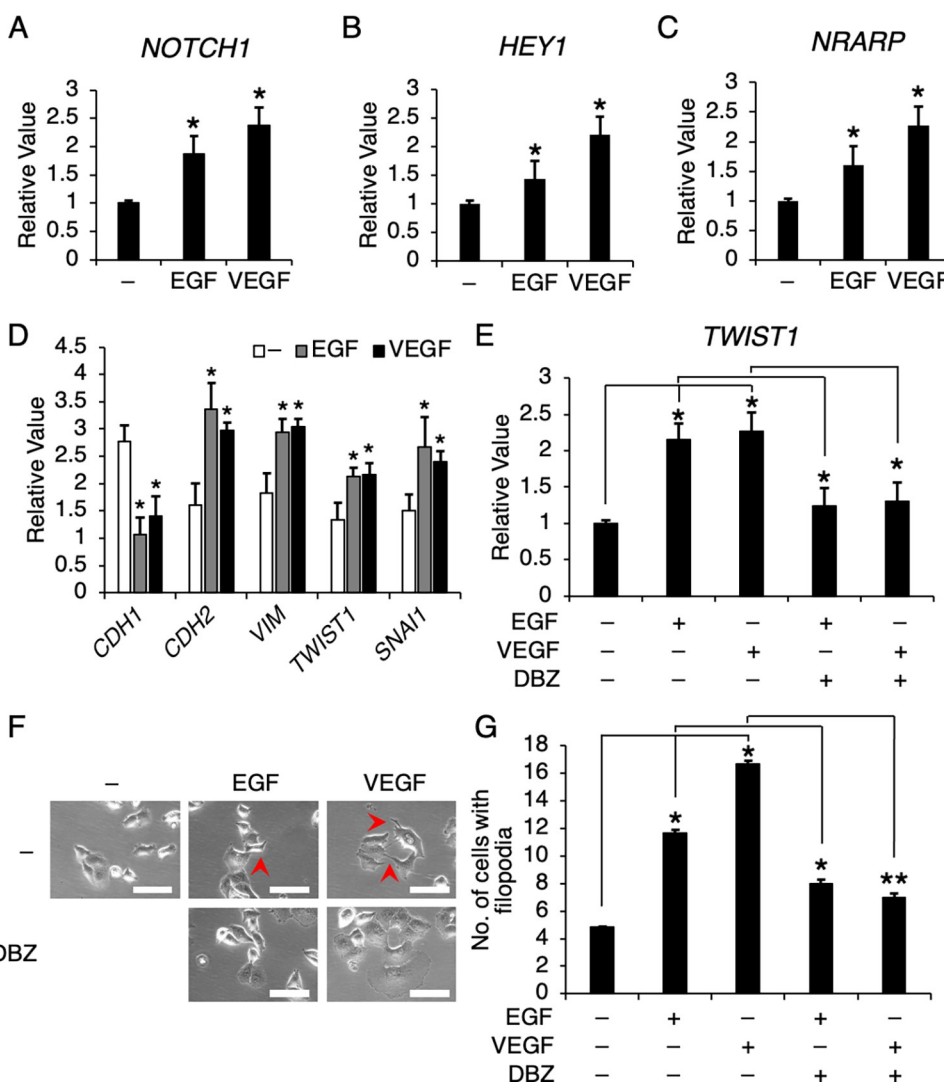

**Fig 5. EGF and VEGF increased Notch activity, which promoted epithelial-to-mesenchymal transition and VM morphogenesis in PDAC cells.** The expression of (A) *NOTCH1* and Notch target genes (B) *HEY1* and (C) *NRARP* were determined in PANC-1 cells treated with EGF or VEGF. (D) The expression levels of an epithelial gene *CDH1* and mesenchymal genes, including *CDH2*, *VIM*, *TWIST1*, and *SNAI1* were determined in PANC-1 cells treated with EGF or VEGF. Notch signaling was inhibited by gamma-secretase inhibitor DBZ at 1 nM, and (E) *TWIST1* expression levels and (F-G) number of cells with filopodia was determined and analyzed in PANC-1. Red arrows, filopodia. Scale bars, 100 μm. Data presented ± S.D. *P value < 0.01, **P value < 0.001 (n = 4–5).

lower in MIA-PaCa-2 when compared to PANC-1 (S6 Fig), which could explain the minimal response upon EGF and VEGF treatment. It is also likely that the regulation of Notch genes is associated with the activation capacity of VM formation in cancer cells, which represents the future direction of our study. In addition, many of these steps, such as cell migration, invasion, and ECM degradation, are also required for cancer progression. This explains why VM formation is often found to be associated with cancer invasiveness. One of the important pathways involved in cancer progression and metastasis is EMT, which has also been implicated in VM capacity in breast cancer cells [30]. EMT regulators were shown to promote the growth and metastasis of PDAC. Nearly 80% of PDAC specimens showed moderate to strong Snail expression, which was inversely correlated with E-cadherin expression [31]. Inflammation also plays

a significant role in PDAC, and NF-kB has been demonstrated to increase both EMT and cancer cell invasion [32,33]. Previous studies have demonstrated that Notch signaling plays an important role in activating EMT during normal development and oncogenic transformation [34–36]. Our data confirmed that SNAI1 and TWIST1 were regulated by Notch signaling and may be involved in cell morphogenesis and tube-like structure formation. And, PDAC cells cultured alone or in Matrigel adjusted their VM capacity in response to angiogenic conditions, suggesting that the VM-forming process in PDAC is likely to be cell autonomous. The utility of our data can be in potentially choosing the appropriate therapeutic strategies for PDAC that would lead to the reduction of blood supply and, consequently, tumor growth by targeting both endothelial cells and cancer cells at the same time. Better understanding the nature of VM and PDAC development can open new avenues for evaluating the impact of anti-cancer and anti-angiogenic therapies.

## Supporting information

**S1 Fig. EGF and VEGF have no effect on VEGFR-2 expression in PDAC cells.** HUVEC and PANC-1 were treated with 20 ng/ml of either EGF or VEGF and cultured for 48 hours. The expression of *VEGFR2* was determined by quantitative RT-PCR. Data presented ± S.D. (n = 3).
(TIF)

**S2 Fig. EGF and VEGF did not induce the expression of Notch downstream target *HES1*.** PANC-1 cells were treated with 20 ng/ml of either EGF or VEGF and cultured for 48 hours. The expression of *HES1* was determined by quantitative RT-PCR. Data presented ± S.D. (n = 3).
(TIF)

**S3 Fig. Inhibition of Notch activity by the gamma-secretase inhibitor DBZ inhibited expression of Notch targets *HEY1*, *HEYL*, and *HES1* and decreased cell viability in PANC-1.** (A) PANC-1 cells were treated with 0.1 nM, 1 nM, 10 nM, 100 nM, and 1,000 nM DBZ and cultured for 48 hours. Cell viability was not affected at 0.1 and 1 nM DBZ but significantly decreased at higher concentrations. Data presented ± S.D. * P Value < 0.001 (n = 3). (B-D) The expression of *HEY1*, *HEYL*, and *HES1* was determined by quantitative RT-PCR. Data presented ± S.D. *P Value < 0.01, **P Value < 0.001 (n = 3).
(TIF)

**S4 Fig. Inhibition of Notch activity by the gamma-secretase inhibitor DBZ abolished the upregulation of *SNAI1* induced by EGF and VEGF.** PANC-1 cells were treated with 20 ng/ml of either EGF or VEGF or 1 nM DBZ and cultured for 48 hours. The expression of *SNAI1* was determined by quantitative RT-PCR. Data presented ± S.D. * P Value < 0.01 (n = 3).
(TIF)

**S5 Fig. Inhibition of Notch activity by the gamma-secretase inhibitor DBZ reduced in vitro endothelial-like PDAC cell morphogenesis in Matrigel.** PANC-1 cells were seeded in Matrigel and treated with 20 ng/ml of either EGF or VEGF and cultured for 24 hours. Network formation was quantitated by Wimasis Image Analysis. Data presented mean number of total tubes ± S.D. (n = 3).
(TIF)

**S6 Fig. Expression of Notch receptors, NOTCH1 and NOTCH2 and their downstream targets, HEY1 and HEYL were higher in PANC-1 than MIA-PaCa-2.** Both cells were cultured up to 80–90% confluency before analysis. Gene expression was determined by quantitative

RT-PCR. Data presented ± S.D. *P Value < 0.05, ** P Value < 0.01 (n = 3).
(TIF)

## Acknowledgments

The authors thank the Central Instrument Facility and the Center of Nanoimaging, Faculty of Science, Mahidol University for technical assistance.

## Author Contributions

**Conceptualization:** Nontawat Benjakul, Thaned Kangsamaksin.

**Data curation:** Nontawat Benjakul, Nattapa Prakobphol, Komgrid Charngkaew, Thaned Kangsamaksin.

**Formal analysis:** Nontawat Benjakul, Nattapa Prakobphol, Komgrid Charngkaew, Thaned Kangsamaksin.

**Funding acquisition:** Thaned Kangsamaksin.

**Investigation:** Nontawat Benjakul, Nattapa Prakobphol, Chayada Tangshewinsirikul, Wirada Dulyaphat, Komgrid Charngkaew, Thaned Kangsamaksin.

**Methodology:** Nontawat Benjakul, Nattapa Prakobphol, Komgrid Charngkaew, Thaned Kangsamaksin.

**Project administration:** Thaned Kangsamaksin.

**Resources:** Nontawat Benjakul, Nattapa Prakobphol, Chayada Tangshewinsirikul, Wirada Dulyaphat, Jisnuson Svasti, Komgrid Charngkaew, Thaned Kangsamaksin.

**Software:** Nontawat Benjakul, Thaned Kangsamaksin.

**Supervision:** Komgrid Charngkaew, Thaned Kangsamaksin.

**Validation:** Nontawat Benjakul, Komgrid Charngkaew, Thaned Kangsamaksin.

**Visualization:** Komgrid Charngkaew, Thaned Kangsamaksin.

**Writing – original draft:** Nontawat Benjakul, Chayada Tangshewinsirikul, Wirada Dulyaphat, Jisnuson Svasti, Komgrid Charngkaew, Thaned Kangsamaksin.

**Writing – review & editing:** Nattapa Prakobphol, Thaned Kangsamaksin.

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
