## [Decision Letter · Decision Letter 0]

21 Sep 2022

PONE-D-22-23184Notch signaling regulates vasculogenic mimicry and promotes cell morphogenesis via the epithelial-to-mesenchymal transition in pancreatic ductal adenocarcinomaPLOS ONE

Dear Dr. Kangsamaksin,

Thank you for submitting your manuscript to PLOS ONE. After careful consideration, we feel that it has merit but does not fully meet PLOS ONE’s publication criteria as it currently stands. Therefore, we invite you to submit a revised version of the manuscript that addresses the points raised during the review process.

We look forward to receiving your revised manuscript.

Kind regards,

Shuai Ren

Academic Editor

PLOS ONE

Journal Requirements:

"This research has received funding support from the NSRF via the Program

Management Unit for Human Resources & Institutional Development, Research and

Innovation (Grant Number B05F640133), National Research Council of Thailand (Grant

Number NRCT5-RSA63015-11), and Mahidol University (T.K.) and the Department of

Pathology, Faculty of Medicine Siriraj Hospital, Mahidol University (N.B. and K.C.)."

"This research has received funding support from the NSRF via the Program Management Unit for Human Resources & Institutional Development, Research and Innovation (Grant Number B05F640133), National Research Council of Thailand (Grant Number NRCT5-RSA63015-11), and Mahidol University (T.K.) and the Department of Pathology, Faculty of Medicine Siriraj Hospital, Mahidol University (N.B. and K.C.). The details of the funders can be found at (1) https://www.nxpo.or.th/B, (2) https://www.nrct.go.th, and (3) https://www.mahidol.ac.th. The funders had no role in study design, data collection and analysis, decision to publish, or preparation of the manuscript."

Reviewers' comments:

Reviewer's Responses to Questions

**Comments to the Author**

1. Is the manuscript technically sound, and do the data support the conclusions?

Reviewer #1: Yes

Reviewer #2: Partly

2. Has the statistical analysis been performed appropriately and rigorously? 

Reviewer #1: Yes

Reviewer #2: Yes

3. Have the authors made all data underlying the findings in their manuscript fully available?

Reviewer #1: Yes

Reviewer #2: Yes

4. Is the manuscript presented in an intelligible fashion and written in standard English?

Reviewer #1: Yes

Reviewer #2: Yes

5. Review Comments to the Author

Reviewer #1: Angiogenesis can arise in a variety of forms during cancer, namely sprouting angiogenesis, glomeruloid microvascular proliferation and vessel co-option. Emerging studies have shown that a few tumors can grow without need of angiogenesis even in hypoxic conditions, while other tumors display both angiogenic and non-angiogenic regions. Vasculogenic mimicry refers to the ability of cancer cells to organize themselves into vascular-like structures for the obtention of nutrients and oxygen independently of normal blood vessels or angiogenesis, recurrently by VE-Cadherin action in cancer cells (PMID: 29786069, PMID: 30816337).

The original article has been well-conducted. However, this reviewer has several points need to be amended:

1.- VE-Cadherin expression: the author only showed a VE-Cadherin mRNA expression in vitro experiments, but the phosphorylation of VE-Cadherin Y658 are correlation with VM formation. The author should be Y658 status in this experiments. At least describe with more emphasis this role.

Reviewer #2: In this manuscript, Nontawat Benjaku et al. describe that NOTCH activity is associated with VM formation. They found that in human pancreatic ductal adenocarcinoma (PDAC), the area of connective tissue hyperplasia is basally vascularized, and that VM volume is positively correlated with tumor size, NOTCH expression, and nuclear localization.

NOTCH, as an important pathway, is closely involved in the regulation of cell fate in development and tissue homeostasis, however, it has been shown that DLL4 of NOTCH signaling is also required for proper vascular growth. The fact that this manuscript finds basal vascularization in the connective tissue region strongly supports the positive correlation of VM with NOTCH expression and nuclear localization. However, I think there is insufficient experimental evidence to reveal that Notch signaling regulates angiogenic mimicry through epithelial-mesenchymal transition in pancreatic ductal adenocarcinoma. This and several critiques listed below greatly reduced my enthusiasm.

Major critiques:

1. It is far-fetched that notch1 is localized in the nucleus in the strong VM formation region, and notch1 in the weak VM formation region is localized in the cytoplasm only by IHC experiments (such as fig3A). Could high expression of notch1 in the cytoplasm induce VM production? They should use more experiments such as IF etc. to further prove this conclusion.

2. The experiment that EGF and VEGF promote cell morphological transformation proves that EGF and VEGF can induce the antennae production of PANC- and MIA-PaCa-2 cells, but this experiment does not have high and low dose settings (such as fig4A). It can promote the expression of notch1, so is this high expression of notch1 a high expression in the nucleus (such as fig5A) to the previous paragraph?

3. I think the experimental design of EGF and VEGF promoting cell morphological transformation is not rigorous enough. If notch1 is knocked out, will the stimulation of PANC- and MIA-PaCa-2 cells by adding EGF and VEGF also promote angiogenesis? Is the strength of promoting angiogenesis the same as the group with normal expression of Notch1?

4. The relevant design of notch1 promoting EMT is not rigorous enough. The experimental results confirmed that EGF and VEGF promoted EMT, but did not prove the effect of notch. EGF and VEGF promoted EMT and EGF and VEGF can promote the expression of notch1. Didn't let me find the relevant logic.

6. PLOS authors have the option to publish the peer review history of their article (what does this mean?). If published, this will include your full peer review and any attached files.

Reviewer #1: No

Reviewer #2: No

---

## [Author Response · Author response to Decision Letter 0]

21 Nov 2022

Dear Editor-in-Chief of PLOS ONE:

We here submitted our revised manuscript entitled, “Notch signaling regulates vasculogenic mimicry and promotes cell morphogenesis and the epithelial-to-mesenchymal transition in pancreatic ductal adenocarcinoma” for consideration as a research article in PLOS ONE. We thank the Editor and Reviewers for recognizing the importance of our study.

To address the Editor and Reviewers’ comments, we have revised our manuscript. Page and line numbers have been added to facilitate the modifications. Our responses to specific comments follow.

Responses to the Editor’s Comments on Journal Requirements:

1. We have confirmed that the manuscript meets PLOS ONE’s style requirements as instructed.

2. The additional statement “The written consent was obtained from all participants prior to tissue analysis” has been added to the Tissue Preparation part of the Materials and Methods section on Page 7.

3. On Page 19, we have removed the funding-related information from the manuscript as instructed. The current funding statement is correct.

4. We have included the data that were not previously shown in the manuscript on Pages 16 and 18 as Supplementary Figures 3 and 6, respectively.

Additional Responses to the Editor

1. We propose to modify the title of the manuscript to “Notch signaling regulates vasculogenic mimicry and promotes cell morphogenesis and the epithelial-to-mesenchymal transition in pancreatic ductal adenocarcinoma.” We have changed the word “via” to “and” in the title. This modification is to address the comments (point 4) from Reviewer 2.

2. We have added one author, Nattapa Prakobphol, to the manuscript for the work that she contributed during the manuscript revision in both conducting additional experiments and writing revised manuscript.

Responses to Comments from Reviewer 1

1. VE-Cadherin expression: the author only showed a VE-Cadherin mRNA expression in vitro experiments, but the phosphorylation of VE-Cadherin Y658 are correlation with VM formation. The author should be Y658 status in these experiments. At least describe with more emphasis this role.

We strongly agree that the role of VE-cadherin should be further explored regarding its role in VM formation. However, it is beyond the scope of our current study at the moment and we are planning to include the function of VE-cadherin, including its expression regulation and activation profiles in our future investigations. We agree with the Reviewer’s comment and have included the discussion of VE-cadherin Y658 and Y731 with references in the Conclusion section on Page 17 and as References 26, 27 and 28.

Responses to Comments from Reviewer 2

1. It is far-fetched that notch1 is localized in the nucleus in the strong VM formation region, and notch1 in the weak VM formation region is localized in the cytoplasm only by IHC experiments (such as fig3A). Could high expression of notch1 in the cytoplasm induce VM production? They should use more experiments such as IF etc. to further prove this conclusion.

First, while the immunohistochemical analysis of Notch localization in the nucleus is a well-established indicator of Notch activity in tissue samples, our results in fact suggested a strong correlation of Notch activity and VM formation in PDAC tissues. In our point of view, Notch immunohistochemistry proved to be a more viable technique than immunofluorescence in order to investigate Notch localization and, at the same time, to identify VM structures, which requires PAS and H&E staining. Second, we also asked a similar question whether VM formation required Notch activity?” This is the reason why we attempted to use an in vitro model to address the question whether Notch signaling plays an essential role in inducing VM structures. Our results indicated that inhibition of Notch signaling in PANC-1 effectively abolished several key characteristics of VM structures, including filopodia formation and cellular branching that were induced by EGF and VEGF. 

2. The experiment that EGF and VEGF promote cell morphological transformation proves that EGF and VEGF can induce the antennae production of PANC- and MIA-PaCa-2 cells, but this experiment does not have high and low dose settings (such as fig4A). It can promote the expression of notch1, so is this high expression of notch1 a high expression in the nucleus (such as fig5A) to the previous paragraph?

First, the reason why we selected only 20 ng/ml of EGF and VEGF in our study was that the same concentration was also used in endothelial cell culture, making it relevant to study the process that was assumed to replace normal angiogenesis. Moreover, we have tested several concentrations (2, 10, 20, 50 ng/ml) of EGF and VEGF in the past and found that we observed effects of cell morphology change only when we introduced 20 ng/ml or higher of EGF or VEGF, and the higher concentrations did not significantly increase the level of cell morphogical impact. Second, EGF and VEGF treatments increase Notch activity as shown by the upregulation of Notch targets, HEY1 and NRARP. Thus, with this evidence we can infer that Notch is activated and downstream events are induced, in a similar fashion to when Notch is translocated into the nucleus to upregulate target genes. 

3. I think the experimental design of EGF and VEGF promoting cell morphological transformation is not rigorous enough. If notch1 is knocked out, will the stimulation of PANC- and MIA-PaCa-2 cells by adding EGF and VEGF also promote angiogenesis? Is the strength of promoting angiogenesis the same as the group with normal expression of Notch1?

We asked a very similar question to the reviewer about the importance of NOTCH1 in the setting of EGF and VEGF-induced VM formation. However, we believe that NOTCH1 is also important for other characteristics of PDAC formation and progression. There, we addressed the question by using a well-established gamma-secretase inhibitor DBZ in order to inhibit Notch activation, rather than a complete knockout of NOTCH1. We have shown that DBZ effectively inhibited Notch activity and abolished the effects of EGF- or VEGF-induced cell morphological changes and EMT-related gene expression induction and added the data in Supplementary Figure 3. Our data provide evidence that EGF and VEGF promote VM formation, which requires Notch activity.

4. The relevant design of notch1 promoting EMT is not rigorous enough. The experimental results confirmed that EGF and VEGF promoted EMT, but did not prove the effect of notch. EGF and VEGF promoted EMT and EGF and VEGF can promote the expression of notch1. Didn't let me find the relevant logic.

We made the conclusion about the role of Notch signaling from the data using a gamma-secretase inhibitor, which effectively abolished the effects of EGF and VEGF on the induction of EMT in PDAC. Additionally, there were several previous studies that support the concept of Notch promoting EMT, including in cardiac development and oncogenic transformation (Timmerman LA, et al., 2004), squamous cell carcinoma (Natsuizaka M, et al., 2017), and TGFbeta-induced retinal fibrosis (Sheng W, et al., 2022). We have added the statement, “Previous studies have demonstrated that Notch signaling plays an important role in activating EMT during normal development and oncogenic transformation” in the Conclusions section on Page 18 (References 

However, we agree that our data were not strong enough to indicate that EMT was the process that was activated to induce VM formation in PDAC. Therefore, we propose to modify the title of the manuscript to “Notch signaling regulates vasculogenic mimicry and promotes cell morphogenesis and the epithelial-to-mesenchymal transition in pancreatic ductal adenocarcinoma.” We have changed the word “via” to “and” in the title.

Thank you for your kind consideration.

Yours Sincerely,

Thaned Kangsamaksin

---

## [Decision Letter · Decision Letter 1]

29 Nov 2022

Notch signaling regulates vasculogenic mimicry and promotes cell morphogenesis and the epithelial-to-mesenchymal transition in pancreatic ductal adenocarcinoma

PONE-D-22-23184R1

Dear Dr. Kangsamaksin,

We’re pleased to inform you that your manuscript has been judged scientifically suitable for publication and will be formally accepted for publication once it meets all outstanding technical requirements.

Kind regards,

Shuai Ren

Academic Editor

PLOS ONE

Additional Editor Comments (optional):

Congratulations on the good work! The paper is ready for publication now.

Reviewers' comments:

Reviewer's Responses to Questions

**Comments to the Author**

1. If the authors have adequately addressed your comments raised in a previous round of review and you feel that this manuscript is now acceptable for publication, you may indicate that here to bypass the “Comments to the Author” section, enter your conflict of interest statement in the “Confidential to Editor” section, and submit your "Accept" recommendation.

Reviewer #1: All comments have been addressed

Reviewer #2: All comments have been addressed

2. Is the manuscript technically sound, and do the data support the conclusions?

Reviewer #1: Yes

Reviewer #2: Yes

3. Has the statistical analysis been performed appropriately and rigorously? 

Reviewer #1: Yes

Reviewer #2: Yes

4. Have the authors made all data underlying the findings in their manuscript fully available?

Reviewer #1: Yes

Reviewer #2: Yes

5. Is the manuscript presented in an intelligible fashion and written in standard English?

Reviewer #1: Yes

Reviewer #2: Yes

6. Review Comments to the Author

Reviewer #1: (No Response)

Reviewer #2: (No Response)

7. PLOS authors have the option to publish the peer review history of their article (what does this mean?). If published, this will include your full peer review and any attached files.

Reviewer #1: **Yes: **Daniel Delgado-Bellido

Reviewer #2: No

---

## [Editor Report · Acceptance letter]

14 Dec 2022

PONE-D-22-23184R1 

Notch signaling regulates vasculogenic mimicry and promotes cell morphogenesis and the epithelial-to-mesenchymal transition in pancreatic ductal adenocarcinoma 

Dear Dr. Kangsamaksin:

I'm pleased to inform you that your manuscript has been deemed suitable for publication in PLOS ONE. Congratulations! Your manuscript is now with our production department. 

Kind regards, 

on behalf of

Dr. Shuai Ren 

Academic Editor

PLOS ONE